# Towards the Monitoring of Functional Status in a Free-Living Environment for People with Hip or Knee Osteoarthritis: Design and Evaluation of the JOLO Blended Care App

**DOI:** 10.3390/s20236967

**Published:** 2020-12-05

**Authors:** Jill Emmerzaal, Arne De Brabandere, Yves Vanrompay, Julie Vranken, Valerie Storms, Liesbet De Baets, Kristoff Corten, Jesse Davis, Ilse Jonkers, Benedicte Vanwanseele, Annick Timmermans

**Affiliations:** 1Human Movement Biomechanics Research Group, Department of Movement Sciences, KU Leuven, 3000 Leuven, Belgium; ilse.jonkers@kuleuven.be (I.J.); benedicte.vanwanseele@kuleuven.be (B.V.); 2REVAL Rehabilitation Research, Hasselt University, 3590 Diepenbeek, Belgium; liesbet.debaets@uhasselt.be (L.D.B.); Annick.Timmermans@uhasselt.be (A.T.); 3Declarative Languages and Artificial Intelligence Group, Department of Computer Science, KU Leuven, 3000 Leuven, Belgium; arne.debrabandere@kuleuven.be (A.D.B.); jesse.davis@kuleuven.be (J.D.); 4Instituut voor Mobiliteit, Hasselt University, 3590 Diepenbeek, Belgium; yves.vanrompay@uhasselt.be; 5Mobile Health Unit, Faculty of Medicine and Life Sciences, Hasselt University, 3590 Diepenbeek, Belgium; julie.vranken@uhasselt.be (J.V.); valerie.storms@uhasselt.be (V.S.); 6Department of Orthopedics and Future Health, Ziekenhuis Oost-Limburg, 3600 Genk, Belgium; Kristoff.Corten@zol.be

**Keywords:** app development, usability, osteoarthritis, blended care, interaction design

## Abstract

(1) Background: Joint loading is an important parameter in patients with osteoarthritis (OA). However, calculating joint loading relies on the performance of an extensive biomechanical analysis, which is not possible to do in a free-living situation. We propose the concept and design of a novel blended-care app called JOLO (Joint Load) that combines free-living information on activity with lab-based measures of joint loading in order to estimate a subject’s functional status. (2) Method: We used an iterative design process to evaluate the usability of the JOLO app through questionnaires. The user interfaces that resulted from the iterations are described and provide a concept for feedback on functional status. (3) Results: In total, 44 people (20 people with OA and 24 health-care providers) participated in the testing of the JOLO app. OA patients rated the latest version of the JOLO app as moderately useful. Therapists were predominantly positive; however, their intention to use JOLO was low due to technological issues. (4) Conclusion: We can conclude that JOLO is promising, but further technological improvements concerning activity recognition, the development of personalized joint loading predictions and a more comfortable means to carry the device are needed to facilitate its integration as a blended-care program.

## 1. Introduction

Osteoarthritis (OA) is one of the main musculoskeletal disorders that causes disability in people over 60 years of age [1]. Symptomatic OA causes pain, joint stiffness, swelling and difficulties performing activities in daily life [1]. The joint that is most often affected is the knee, followed by the hip. These are two large, weight-bearing joints of the lower extremities that are crucial for locomotion [1]. OA is a multifactorial disease in which mechanical factors play an essential role [2,3,4,5,6,7]. The conventional treatment of OA patients consists of education, the promotion of an active and healthy lifestyle, medication, exercise therapy [8] and—as a last resort—a total joint arthroplasty [9]. Previous research has shown that inactivity is a predictor for increased OA symptoms and poor general health, which indicates that people with OA should be more active [10]. It has been shown that exercise therapy and sports reduce pain and increase muscle strength, joint stability, physical function and overall health [11,12]. Thus, there should be a balance between the imposed loading on the joint and the person’s ability to carry a load [12]. Furthermore, each daily activity imposes a different mechanical load on joints [13,14]. Therefore, it seems relevant for health-care practitioners to gain insights into and monitor a patient’s functional status—i.e., activity profile—in combination with their joint loading in a free-living environment. State-of-the-art biomechanical modeling tools based on integrated 3D motion capture in a laboratory setting can quantify joint loading. However, access to these tools and the need for expert operators limits the accessibility of measuring joint loading in a clinical setting.

Over the last decade, the use of mobile technology has increased exponentially [15], and smartphones have become an integral part of people’s lives [16]. Although interesting work has been presented on the use of technology to support the elderly in performing exercises, such as the Nintendo Wii or Microsoft Kinect (e.g., [17]), our paper focuses on the use of smartphones or Inertial Measurement Units (IMU) to enhance the uptake of physical activities in daily life. Smartphones allow users to engage with information in any environment at any time. Furthermore, they are equipped with technologies such as a tri-axial accelerometer, tri-axial gyroscope and a magnetometer (IMUs), and they can therefore monitor physical activity by accessing these sensors and combining their information in a smartphone application [18]. Furthermore, the graphical interface of applications (apps) allows tailored feedback, social interaction, personalized goal setting and self-monitoring [19,20], which might help with the commencement and maintenance of physical activity [21].

However, starting and maintaining physical activity in order to increase healthy behavior is challenging [21,22]. Results show that the positive effects of exercise in people with OA only last as long as exercise is continued [22]. Unfortunately, reasons such as forgetfulness, boredom, lack of enjoyment and lack of confidence are cited as reasons to stop exercising [22]. Interventions that promote exercise using only app-based feedback peak in the first few months, and adherence to the exercise plan dwindles over time [18]. Thus, while smartphone apps increase self-monitoring through direct feedback and goal setting, they lack the social support and controlling function provided by a health-care professional [23]. Considering that supervision facilitates the adherence to exercise interventions [22], an app-based approach alone might not be sufficient. Furthermore, OA patients highlighted that contact with a therapist [20] or peers [23] throughout the intervention is essential for motivation and to receive corrective feedback and reassurance. Moreover, older adults (with a lack of motivation to use technology) and people with comorbidities (who experience more physical discomfort during activities) might need the personal approach of face-to-face therapy sessions [24] as opposed to solely relying on app-based interventions. In such cases, a blended care approach (the combination of mobile technology with conventional therapy) might be more useful. Over the last decade, more researchers have become focused on applying technology in a rehabilitation setting to enhance physical activity in people with OA and have indicated success in that regard [24,25]. However, to the best of our knowledge, no app to date incorporates mechanical loading on the joint (i.e., joint loading in terms of contact forces) during activities of daily life. Therefore, we aim to add an estimate of the cumulative joint loading based on the recognition of the performed activity during daily life for people suffering from hip or knee OA.

We propose the concept and design of a novel blended care app called JOLO (Joint Load) that combines free-living information on activity with lab-based measures of joint loading to estimate functional status. As such, JOLO aims to provide a better overview of a patient’s functional status. Subsequently, the patient’s rehabilitation strategies can be further optimized. For that purpose, we deploy JOLO on the readily available smartphones of patients and provide a clinical dashboard for therapists (accessible on their PC or smartphone). Within this study, we report the development and the evaluation of the usability, feasibility and credibility of the JOLO app.

## 2. Materials and Methods

### 2.1. System Design and Developments

JOLO contains four modules, (1) the JOLO framework, (2) activity recognition, (3) functional status estimation and (4) personalized goals. The design for the high-fidelity prototypes of the smartphone app and the clinical dashboard are shown in Figure 1 and Figure 2 respectively.

#### 2.1.1. Module 1: JOLO Framework: Technologies and Architecture

The JOLO framework is an integrated solution that provides health-care professionals and patients with the necessary tools to follow up and manage rehabilitation. The patient app is both a data provider and data presenter: accelerometer data are gathered by the app and processed in the backend, and the resulting activities and joint load values are shown by the app to the patient. The clinical dashboard is used only by health-care professionals to set activity and joint load targets for patients and to manage and follow up on the rehabilitation progress. Both frontends (app and clinical dashboard) rely on the backend for data processing and storage.

The JOLO framework uses the following technologies (Figure 3):PostGreSQL: Processed patient activities are stored in the JOLO PostGreSQL database. Patient movement data are pseudonymized and stored in an encrypted way in the JOLO database compliant to GDPR rules and guidelines. A separate database contains patient personal information and is only accessible by the authorized researchers in a virtual private cloud.Laravel: The postprocessing backend and clinical dashboard are built using the Laravel PHP framework, which allows for a modular and flexible state-of-the-art approach to software architecture.Python: The toolbox that infers patient activities from sensor data is written as a Python script.Kotlin: The JOLO smartphone app for Android is written in Kotlin.

Upon subject inclusion, the responsible researcher/therapist registers the patient on the clinical dashboard and the patient’s personal information is stored in the Dharma database in a pseudonymized manner. Subsequently, the patient can log into the smartphone app using their credentials. The smartphone registers raw tri-axial accelerometer data at 50 Hz. This data are merged in csv files, zipped and sent to the Laravel backend every 5 min if network connectivity (i.e., WIFI) is available. In case the network is unavailable, the sensor data are stored locally on the smartphone and sent as soon as a network connection is active again. To send the data to the online database, the app performs a POST REST API call on the backend. In addition, the data are stored on the Amazon S3 service for secure backup. The activity toolbox is notified each time a chunk of data arrives, fetches the zip and processes the data to create a file which contains the activity predictions (see Section 2.1.2 Module 2: Activity recognition). These predictions are further processed by the Laravel backend and stored in the JOLO database. Patient data are visualized in an aggregated form both on the smartphone app for the patient and on the clinical dashboard for the therapists, for which a REST API is exposed over HTTPS to both frontends to retrieve the data.

#### 2.1.2. Module 2: Activity Recognition

The second module of JOLO detects the patient activity by applying a machine learning model to the accelerometer data collected while the patient wears the phone in a hip bag. The model can identify seven activities: walking, ascending stairs, descending stairs, sit-to-stand, stand-to-sit, jogging and cycling. There are several publicly available datasets and models used to detect human activities. The most similar dataset is the HAR dataset [26]. This dataset includes a similar set of activities (i.e., walking, ascending/descending stairs and sitting/standing, but not cycling or running) and the IMU data was also collected using a mobile phone in a hip bag. However, after training a model on the HAR dataset, we found that the accuracy for our data was low, which was probably due to a slightly different position of the hip bag. Therefore, we developed our own activity recognition model, as none of the existing approaches exactly matched the sensor position used in our study.

We used machine learning to train an activity recognition model that correlates the accelerometer measurements to each activity. To do so, we collected 14,597 s of labeled data from 17 healthy participants (aged between 23 and 66). We standardized data collection by using a standardized movement protocol and an identical smartphone (Samsung Galaxy J5) and by providing an identical hip bag to all participants with instructions on how to wear them. The participants were instructed to perform a series of multiple repetitions of each exercise; i.e., bouts of walking, ascending and descending stairs, sit-stand transitions, jogging and cycling. Table 1 shows the duration of the collected data for each activity. During each exercise repetition, the phone collected the tri-axial accelerometer data; i.e., three synchronized time series $a_{x}$, $a_{y}$, $a_{z}$, sampled at 50 Hz. (We collected the data from the mobile phone by recording the data from the TYPE_ACCELEROMETER sensor type of the Android system. The signals returned by the Android system were corrected for drift in the sensor measurements, as shown in the API documentation (https://developer.android.com/guide/topics/sensors/sensors_motion).) Note that we only used the accelerometer and no other IMU sensors from the phone. While many models use only accelerometer data [27], it is possible to obtain a more accurate model when also including the gyroscope and magnetometer. However, due to the relatively small dataset size, we decided not to include these to reduce the risk of overfitting. We trained the model based on the collected data using the following steps:Preprocessing: Instead of using the raw tri-axial accelerometer data ($a_{x}$, $a_{y}$, $a_{z}$), we took the absolute value along each axis ($|a_{x}|$, $|a_{y}|$, $|a_{z}|$) in order to correct for different orientations of the mobile phone. Consequently, the user could position the phone in any orientation in the hip bag.Segmentation: In order to detect the activity performed at each second, we followed a sliding window approach with a step size of one second (50 samples) and a size of three seconds (150 samples). We labeled each window with the activity performed by the participant.Feature extraction: We transformed the preprocessed accelerometer measurements ($|a_{x}|$, $|a_{y}|$, $|a_{z}|$) to a feature representation to reduce the dimensionality of the data and reduce the risk of overfitting. Using the tsfresh Python package [28], we extracted the same set of 794 features from each acceleration signal. Each feature summarized the 150 samples of one signal as a single value. Examples of such features are the mean acceleration, the number of zero crossings, Fourier transform coefficients, etc. Using the same package, we selected a set of 98 relevant features from the 3×794 features.Training: In order to correlate the features to the activity, we trained a gradient-boosted decision tree ensemble model that predicted the activity based on the features extracted for one window. We used the CatBoost Python package [29] to train the model.

To apply the model to the data continuously collected by the application while a patient was wearing a smartphone, the raw accelerometer data were preprocessed, segmented into windows and transformed into a feature representation. The features were used as inputs to the learned gradient-boosted decision tree ensemble, which predicted one of the seven activities every second.

We evaluated the predictive accuracy of the model using leave-one-subject-out cross-validation. In this cross-validation scheme, the dataset was split multiple times into a training and test set, where the test set contained the data of one subject and the training set contained the data of all other subjects. The model obtained an accuracy of 94.64% using this cross-validation procedure. Table 2 shows the confusion matrix, which provides a more detailed overview of the accuracy for each activity.

While the model shows high accuracy for the collected data, the accuracy may be lower when applied in a free-living environment. The model can only recognize the seven activities it was trained on. In reality, people will perform many other activities beyond these seven, and whenever they do, the model will always make an incorrect prediction. Even when the user performs one of the seven activities, a window can contain two activities when the person transitions from one activity to another. To evaluate the model on free-living data, we analyzed additional data from realistic scenarios, collected from nine healthy participants. The participants performed sequences of activities that resembled daily-life situations. We added a postprocessing phase to the model to improve the accuracy for these sequences. The postprocessing steps were as follows:Instead of simply predicting the activity of each window, we also predicted the probability that a person was performing each of the seven activities. The highest probability (i.e., the probability corresponding to the predicted activity) provided an indication about how certain the model was. When this probability was lower than a threshold (δ), we replaced the predicted activity with an “unknown” activity label. For example, if the model predicted “descending stairs” for a certain time window but the probability associated with “descending stairs” was lower than δ, we replaced the predicted label “descending stairs” with the label “unknown”. The model may be uncertain in its predictions when the user performs other activities beyond the seven activities in our dataset, and when the user transitions from one activity to another. Replacing the prediction with “unknown” can reduce the number of incorrect predictions in these cases. To determine δ, we evaluated the accuracy as well as the percentage of data that was not replaced with an “unknown” label. Figure 4 shows the results for this experiment. Based on these results, we set δ to 0.5, where the accuracy improved slightly (from 72.82% to 74.43%) while 96% of the data were still not replaced with an “unknown” label. Taking a higher value for δ would improve the accuracy further, but this would also increase the number of predictions that are replaced by an “unknown” label. This is not desirable, as we may risk large chunks of activities being replaced by “unknown” even though they were detected correctly, albeit with a small probability.For sitting and standing, the model only detects sit–stand transitions. Thus, periods of sitting and standing are always misclassified as one of the seven activities. However, it is easy to detect these periods, as sitting and standing are the only activities in which the phone is not moving. We add a rule that classifies a window as “sitting or standing” when the variance of the $a_{x}$ acceleration is low (smaller than 0.1) and applies the learned model otherwise. Note that we cannot discriminate between sitting and standing, as the user is allowed to wear the phone in any orientation.

In addition to classifying the user’s activity, the application also estimates the number of cycles for each activity; i.e., the number of strides when walking, ascending stairs, descending stairs or running, the number of sit–stand transitions and the number of rotations for cycling. For sit–stand transitions, this is simply the number of windows, as one repetition only takes one window in duration. To estimate the number of cycles for the other activities, we first estimate the cadence; i.e., the number of cycles per second. We follow an approach similar to [30] and find the cadence based on the Fourier transform of the acceleration. Specifically, we compute the discrete Fourier transform of the resultant acceleration ($\sqrt{a_{x}^{2}+a_{y}^{2}+a_{z}^{2}}$) for each three-second window. The estimated cadence is the frequency for which the amplitude of the Fourier transform is maximal. We then take the sum over all seconds to compute the total number of cycles.

#### 2.1.3. Module 3: Functional Status

The third module assesses the subject’s functional status, which consists of estimating the subject’s joint loading and activity profile. Here, we approximate the joint contact force impulse of the hip and the knee joint during each of the detected activities by using joint loading templates.

In another study (the medical ethical committee of the academic hospital Leuven (s-59857) approved this study), we addressed the calculation of the hip and knee contact forces [31]. In short we invited 12 healthy controls (age: 59.7 ± 7 years), 20 people with end-stage hip OA (age: 63.1 ± 6.2 years) and 18 people with end-stage knee OA (age: 65.1 ± 5.1 years) to perform a functional movement protocol at the Movement and Posture Analysis Laboratory (MALL, KU Leuven, Belgium). The functional movement protocol comprised level walking, ascending and descending stairs, sit-to-stand and stand-to-sit transitions, a forward lunge, sideward lunge, single-leg stance and single-leg squat. We used integrated 3D motion capture (Vicon, Oxford Metrics, UK) and ground reaction forces (AMTI, Watertown, MA, USA) from five repetitions of each exercise as the input in a musculoskeletal modeling workflow in OpenSim 3.3 [32] to calculate the resultant joint contact forces of the hip and knee [13,33]. The average joint contact force impulses for hip OA, knee OA or controls expressed in (N/BW)· s for one cycle of an activity (e.g., one walking stride, one stand-up transition) were used to form the joint load templates no-OA, HipOA and KneeOA (Table 3). Since we had a relatively small data set, we were unable to identify more distinct subgroups; thus, a person was assigned to one of these three profiles based on their diagnosis.

In order to make functional status more intuitive and easier to understand, we expressed the functional status in points per day. Points were allocated based on the activities that were performed during the day (Table 3). For example, descending stairs imposes a higher mechanical load than walking, thus more points were assigned per step spent ascending or descending stairs; (i.e., 1.72 points were awarded to the knee joint for each ascended stair in the KneeOA profile) compared to one step during level walking, which was awarded 1 point per cycle for the knee joint in the KneeOA profile. We estimated the functional status of a subject by multiplying the number of steps/repetitions of an activity to the points of that activity divided by the total number of points that the person needed to reach. Reaching 100 points would be considered the standard goal, and that would be equivalent to the cumulative loading of 10,000 steps/day. The app supports the development of a personalized rehabilitation program in terms of a patient’s functional status by allowing the health-care professional to adjust the total number of points in the clinical dashboard to account for individual differences, personal goals and the stage of the patient’s rehabilitation trajectory. Using this point system makes it easier for both patients and therapists to interpret the functional status.

#### 2.1.4. Module 4: Personalized Goals

The fourth module allows the setting personalized goals for each patient regarding activity and functional status. The JOLO app is designed as a blended care program, meaning that the app facilitates the interaction between the patient and the therapist. Furthermore, successful rehabilitation depends on the expectations and goals of the patients; the app needs to account for this. The therapists’ clinical dashboard contains a section in which the therapists, in collaboration with the patient, can indicate which goals need to be accomplished (Figure 2). It is possible to set goals for both the automatically detected activities and functional status as well as for the patient’s other physical activities. The goals can be specified by indicating the duration and frequency of the activity (e.g., 30 min of cycling, five times per week). The patient will receive feedback in the app on their progress towards reaching these goals (Figure 1). Activity goals can only be achieved if they occur in blocks of at least 10 min [34].

### 2.2. Study Design

We used a user-centered iterative design process to procure information about the JOLO app in two different phases; each phase contained one or two iterations (Figure 5). The iterative design process involves a design–test–redesign–retest cycle [35]; therefore, each iteration resulted in invaluable information to redesign and improve the prototype. The first phase for the smartphone app was scenario-based testing in a controlled environment. The second phase was a 7-day field test of the app in a free-living situation. People who suffered from hip or knee OA evaluated the smartphone app. Health-care professionals—i.e., orthopedic surgeons and physiotherapists who are involved in the treatment of people with OA—assessed the clinical dashboard.

We conducted all experiments following the Declaration of Helsinki of 1975, revised in 2013. The local medical ethical committee of Ziekenhuis Oost-Limburg, Genk approved this study (B371201734378). Before the start of the study, participants provided written informed consent.

### 2.3. Study Sample

All OA patients were recruited from Ziekenhuis Oost Limburg (Genk, Belgium). We asked them to participate after their appointment with their orthopedic surgeon scheduled on the same day. People with OA could participate when they met the following inclusion criteria: aged above 50 years old and diagnosed with hip or knee osteoarthritis, intermittent or constant pain (VAS > 2/10), no total hip or knee arthroplasty or >6 weeks after a total hip/knee arthroplasty, able to walk for a minimum of 10 min, no other musculoskeletal or neurological disorders that affected their gait pattern, ability to speak and read the Dutch language and willingness to work with a smartphone. In total, 20 people with either hip or knee OA participated in the testing and evaluation of the smartphone app.

The health-care professionals were a group of orthopedic surgeons from Ziekenhuis Oost Limburg (Genk, Belgium), and physical therapists from the researchers’ network and the Leuvense Kinekring (a local trade union in Leuven, Belgium). Health-care professionals were included when they were a physical therapist or orthopedic surgeon involved in the rehabilitation of people with OA and were willing to work with technology. In total, 24 health-care professionals participated in the testing and evaluation of the clinical dashboard (nine orthopedic surgeons and 15 physical therapists). All participant characteristics can be found in Table 3 and Table 4. At least five participants evaluated each prototype. This participant number was based on the premise that five people are likely sufficient to uncover major design problems and help us improve the usability of the app [35].

### 2.4. Data Collection

#### 2.4.1. Low-Fidelity Prototype

The first phase was creating, evaluating and refining a low-fidelity prototype. The first low-fidelity prototype (Section A.1) was created based on a focus group meeting with physical therapists who were working with people with OA. The testing and refinement of this prototype was done in three iterative design stages separated by two stages of usability testing. This version was a mock-up of the smartphone app and clinical dashboard, both created in Adobe’s Creative Cloud. In these instances, the smartphone app was displayed on a laptop and not a smartphone. Furthermore, because these versions did not have a backend with actual data or functionality, we used a scenario-based method for the participants to interact with the prototype. This meant that the participant needed to use the JOLO app to complete assignments such as the following: “Imagine you want to know if you/your patient reached the goals in the first week of June, which information do you need and where would you find that?” (see Appendix B for all scenarios). After completing the scenarios, we asked participants to fill out the System Usability Scale (SUS) [36]. Furthermore, after attempting all scenarios, the participants indicated which scenarios they could not complete, and where and how we should improve the app to resolve those issues; notes were made of this and implemented in the app before the next test iteration.

#### 2.4.2. High-Fidelity Prototype

After completing the first phase, the second phase was started. This phase entailed the testing and refining of the high-fidelity prototype as designed and described in Module 1 (Figure 1 and Figure 2). The high-fidelity prototype was refined in three iteration design stages divided by two stages of usability testing. To evaluate the first version of the high-fidelity prototype of the smartphone app, we asked the participants to use the app with a scenario-based method using data stored in the app. After completing the scenarios, we asked them to fill out the Usefulness, Satisfaction, and Ease of Use Questionnaire (USE) [37], the Unified Theory of Acceptance and Use of Technology (UTAUT) [38] and the Credibility and Expectancy Questionnaire (CEQ) [39]. For the second version of the high-fidelity prototype, we wanted our participants to field-test the app. Therefore, after an initial test at Ziekenhuis Oost-Limburg, we asked them to take the smartphone with the app and hip bag home and to wear/use it for seven consecutive days. After this field test, we asked them to fill out the USE, UTAUT and CEQ. Again, we obtained invaluable information by asking our participants to provide oral feedback on each iteration regarding where and how to improve the smartphone app. To evaluate the clinical dashboard, we again used a scenario-based method with similar scenarios as for the low-fidelity prototype (Appendix B). After completing the scenarios, we asked participants to fill out the USE [37] and UTAUT [38]. Again, we asked all participants to indicate where and how the app could be improved.

#### 2.4.3. Questionnaire Scoring

The SUS is a short, valid questionnaire that assesses the overall usability of a system [36,40]. SUS scores are calculated using the formula from Brooke (2013) [40]. SUS scores above 50.9 are considered “OK”, above 71.4 are “good” and scores surpassing 85.5 and 90.9 are “excellent” and “the best imaginable” respectively [41]. The USE and the UTAUT are scored on a seven-point Likert scale. The different dimensions of those questionnaires are averaged, and every score above 3.5 is a positive outcome. The anxiety dimension of the UTAUT is an exception because it needs to be below 3.5 to be positive [37,38]. The scores for the CEQ are summed, with a maximum score of 27. Scores above 13.5 are considered positive [39]. Based on the results of these questionnaires and subsequent conversations, we obtained an in-depth view of the participants’ opinions on the usability, feasibility, credibility and problem areas of the app and dashboard.

## 3. Results

### 3.1. Phase I: Low-Fidelity Prototype

The low-fidelity prototype was refined in three iterative design stages separated by two stages of usability testing. The participant characteristics in terms of age, sex and their confidence in technology use are given in Table 4. Confidence in technology use was scored from 0—“no confidence at all”—to 10—“extremely confident”. Prototype I was designed using information from a focus group of end-users. People with OA tested this prototype using a scenario-based method, and the usability of the patient’s app was “poor” and not acceptable (Figure 6) [41]. From the participants’ interaction with the app, we obtained valuable information through oral feedback, and based on that feedback, we learned that we needed to simplify the interaction between the different screens to improve the retrieval of different types of information. For example, when someone wanted to know if the goals were reached, they only got feedback in the form of faces: a positive face indicated reaching the goal, whereas a sad face indicated not reaching the goal. However, this only gave a dichotomous answer and was not satisfying for the OA patients. They also wanted to know the extent to which they reached the goal, or how much they still needed to do to reach it. In order to retrieve that type of information, they needed to go back and forth between the goals and activity screen, which was cumbersome and time-intensive. Therefore, we changed this to a more compact bundle of information in which the information about the minutes of activity and the goal would be displayed together. These changes to the design led to an increase in the median usability score of 160% to a median SUS score of 60 (Figure 6), which is marginally sufficient. In particular, the lower scores improved to a more acceptable level [41]. One of the main issues that remained was the difficulty in understanding the term joint loading; i.e., how to interpret it. After explaining to the participants that 100 points related to 10,000 walking steps, the OA patients responded positively, and when they saw that the fictive data only had 43 points, they indicated that the person needed to be more active during the day. This taught us that an explanation of the point system is necessary in order to make the smartphone app more understandable and usable. To do this, we created a “help” section in the app, where all data, points and symbols were explained in detail.

The usability for the clinical dashboard in the first iteration was already good (Figure 6) [41]. However, one of the main remarks was to simplify the screen interactions. Some scenarios required the health-care professionals to click too many times, which was too time-intensive. This led us, in a similar way to the smartphone prototype, to a design that bundled more information on one page. Thus, instead of having to navigate between the goals and the activity screen, we bundled this information into one graph. When we tested the improved prototype, the median score of the SUS did not improve. The first quartile decreased, which was mainly because two clinicians did not believe that joint loading could be estimated using solely a smartphone. This belief taught us that a clear explanation of the different app modules would be necessary to prevent misconceptions in the interpretation of the functional status in the app. Similarly to the smartphone prototype, we created a “help” section in which the information of the functional activity module was explained.

### 3.2. Phase II: High-Fidelity Prototype

The high-fidelity prototype was refined in three iterative design stages divided by two stages of usability testing. The participant characteristics in terms of age, sex and their confidence in technology use can be found in Table 5. The first iteration was planned as an initial test conducted at Ziekenhuis Oost-Limburg followed by seven days of home use. However, with the first subject during the initial test, we noticed that the data were not uploaded in real-time in order to provide real-time feedback. Therefore, we could not check if the data were successfully uploaded to the server or if the activity recognition was correct. For this reason, we used scenario-based testing for the first version of the high-fidelity prototype using fictive activity and functional status data in the app. The second smartphone prototype was evaluated after seven days of home use. The results of the UTAUT, USE and CEQ can be found in Figure 7 (Table A1), Figure 8 (Table A2), and Figure 9 respectively.

The prototype of the smartphone app tested in a controlled environment (Prototype I) received an overall positive score on usability. Mainly in terms of performance expectancy (the degree to which a person expected the app to help them during their rehabilitation), positive attitudes towards using the app and its credibility in assisting in rehabilitation scored highly. However, due to the delay in the real-time data transfer for the first participant, we learned that the data sharing between the phone and the Dharma database needed to be accelerated. Therefore, the main changes between the first and second prototype were in the backend of the app to accelerate the data transfer between the smartphone and the Dharma database. Because the rest of the scores were high and the subjects gave no added information about changes that needed to be made, not much was changed in terms of the design and screen interactions. However, after the improvements to the data transfer, the smartphone app scored less well after seven days of home use (Prototype II) than during the more standardized setting. These decreases are mainly visible in the performance expectancy (from a 6 to a 2.75) and behavioral intent (from a 5.33 to 2.67; wanting to use the app) from the UTAUT, the usefulness (5.25 to 3.5) and satisfaction (5.14 to 3.14) from the USE, and both credibility (24 to 13) and expectancy (22.8 to 15) from the CEQ. The participants indicated that the app’s activity recognition module was not yet optimal and that it predicted excessive amounts of cycling. The misclassification of cycling and underestimation of stair climbing made the app unconvincing according to the participants. Furthermore, the hip bag used to secure the smartphone was too small for some participants, and the material caused a rash in those individuals. Furthermore, they found it hard to see this app as something other than an activity recognition app that does not work as well as it should and that is uncomfortable to wear for some people.

The clinical dashboard received a variable score from the healthcare professionals. It received good scores for attitudes to the technology, anxiety and the facilitation conditions from the UTAUT, and its ease of use and ease of learning (USE questionnaire). However, the usefulness (USE), performance expectancy and behavioral intent (UTAUT) received low scores. Those scores indicate that the dashboard is easy to use; nevertheless, the health-care professionals are currently unwilling to use it. The health-care professionals indicated that, although they would like to receive information regarding a patient’s functional status based on multiple activities, they have three main concerns that should be addressed prior to implementation; (1) the accuracy of the estimation of the functional status, (2) that it is not suitable for all their patients (only suitable for proficient smartphone users) and (3) the time it takes to use the app.

## 4. Discussion

In this study, we aimed to introduce a novel blended care app called JOLO (Joint Load) for health-care professionals and subjects suffering from OA to optimize their rehabilitation. JOLO combines information from activity recognition in a free-living environment with the joint contact force information collected in a lab-based setting to estimate a patient’s functional status in terms of activity and joint loading in a free-living environment. End-users tested JOLO using an iterative design process in which the end-users provided essential information for the further development of the JOLO app. Based on the results of this study, we can conclude that JOLO is promising but still needs improvement before OA patients and therapists can use it in a rehabilitation setting.

During the last stage of prototype testing, we found that the main issues with the JOLO smartphone app and dashboard were not related to its usability but to its utility. The utility consists of the functionalities of a system—whether it does what it was designed for and whether it facilitates the users’ job [44]. For the smartphone app, the users indicated that some functionalities of the system were not working correctly. The subjects that used JOLO for seven days in a free-living environment indicated that there was an excessive overestimation of the time spent “cycling”. Considering that technical difficulties are in general identified as a vital barrier for the effective use of and satisfaction with technology [45], improving the predictive performance of the activity recognition module should be guaranteed in future prototype versions. Currently, every second of the day is allocated to a specific activity or to “unknown”. To reduce the errors of allocation, it is probably more appropriate to opt for allocation based on a combination of acceleration data and logical reasoning. If, for example, “cycling” is predicted for a short amount of time, followed by “stationary” for a more extended period, it is unlikely that a short cycling bout took place. In such a case, it is more appropriate to change the short amount of time allocated to “cycling” to “stationary”. Furthermore, the training data in our machine learning pipeline were mostly collected from healthy and slightly younger individuals without any musculoskeletal deficits. Previous work has shown that there are significant differences in movement patterns between healthy individuals and those with hip or knee OA [46,47]. Those differences might also induce differences in the acceleration signal, and in turn change the accuracy of the model and result in more misclassifications. This issue should be addressed in future work on the JOLO app by including data from patients with hip or knee OA to train the classification model. Another limitation of the activity recognition pipeline is the limited number of input sensors. To reduce the risk of overfitting with our relatively small dataset, we decided to keep the dimensionality of the input data low by only using the accelerometer. If more training data were made available, future work could use the other IMU sensors in the phone (i.e., gyroscope and magnetometer) to train a more accurate model. Besides the issues related to activity recognition, some participants indicated that the hip bag used to wear the smartphone was too small, and thus for some patients it was uncomfortable to wear the smartphone. An essential aspect of using a device for continuous monitoring is its natural and non-intrusive integration in everyday life [48]. Therefore, wearing an uncomfortable bag might explain some of the lower scores for behavioral intent and satisfaction. A custom made hip bag that is large enough and comfortable to wear might be a better solution than using a commercially available option.

Usefulness and ease of use have been identified as two of the primary parameters facilitating the use of mobile health by healthcare professionals [49]. The health-care professionals indicated that the last prototype of JOLO was easy to use and easy to learn. They were furthermore interested in receiving their patients’ activity data based on multiple activities in a home environment. One of the main issues with physical activity prescription for health-care professionals lies in their inability to monitor their patients [22]. In that sense, using JOLO might facilitate the prescription of physical activity to these patients because it allows for detailed monitoring during daily life. However, the use of JOLO currently costs more time than it saves, and as the health-care practitioners indicated, it might not be suitable for all their patients. Their concerns might be reflected in the spread of the UTAUT and USE scores of the patients (Table A1 and Table A2). This spread might be caused by differences in how confident the participants felt in using technology. Currently, we have treated confidence in technology use as a participant characteristic, similar to age and gender. However, the visual inspection of the scatter plot shows that there could be a correlation between the self-reported confidence in technology use and the attitude and anxiety towards using the JOLO app. Such a correlation would indicate that some people would need more guidance and training before they would become comfortable in using technological products for rehabilitation. Future work should further investigate this relationship.

Moreover, the usefulness of obtaining information and providing feedback on joint loading in a free-living environment is not yet proven. Undesirable mechanical loading plays a critical role in the onset and progression of OA [2,4,50]. However, what constitutes “undesirable” mechanical loading on the joint in a free-living environment is currently unknown. Therefore, providing feedback and information on this parameter is presumptuous. For this reason, the therapists indicated that the app is usable but currently not useful. Therefore, the health-care professionals indicated that, before JOLO can be implemented in a rehabilitation setting, benchmarks for proper joint loading need to be added to the app. Furthermore, the health-care professionals also expressed their concerns regarding the accuracy of the joint loading based on group averages and indicated that more personalized methods should be implemented. Currently, the population average based on lab-based data is used as a joint loading template. However, this will induce an error when estimating the functional status of an individual. The current workflow used to calculate joint contact forces involves an extensive lab-setup and musculoskeletal modeling. It is not feasible to implement such a workflow in a rehabilitation setting to obtain an individual’s joint loading profile. Therefore, alternative methods to calculate or predict joint contact forces should be investigated so that they can be used to obtain a more personalized functional status. Previously, IMUs have been used to estimate ground reaction forces and moments (e.g., [51,52,53]) and even joint contact forces (e.g., [54,55]) using machine learning techniques. Those results show the potential of using IMUs to obtain complex measures outside of the lab. However, more research is needed to reduce the prediction errors on the implementation within a rehabilitation setting and on the implementation of this measure in the JOLO app. Therefore, JOLO can only be implemented in a rehabilitation setting if these issues are simultaneously addressed.

Future work on the JOLO app should investigate feasible methods to estimate joint contact forces in a free-living situation using a combination of IMU sensors and machine learning techniques. These techniques will allow for more personalized joint loading templates and reduce the error in the functional status estimation. Furthermore, benchmarks for healthy joint loading should be investigated so that they can be used as a guideline for therapeutic goal setting. This can be done by exploring the differences in functional status between healthy individuals and individuals with hip or knee OA. Providing benchmarks and testing the effectiveness of the feedback that benchmarks the individual patient performance against healthy individuals or well-recovering peers should prove the usefulness of the JOLO app and facilitate its integration in rehabilitation. Moreover, to ease the natural integration in everyday life, a different carrier for the phone should be found by designing a custom made belt with skin-friendly material that is adjustable to common anthropometric measures. To do this, different types of prototypes, in terms of materials, shapes and sizes, should be created and field-tested. Furthermore, the activity recognition algorithms should be improved to reduce the amount of misclassification, possibly by adding real-life labeled data from hip and knee OA patients to the training data set.

## 5. Conclusions

This paper presented the concept and design of a novel blended care app called JOLO (Joint Load). JOLO combines data from a free-living environment on multiple activities of daily living with lab-based measures of joint loading to estimate the functional status of people with OA. We found that people with OA and health-care professionals in charge of the rehabilitation of people with OA were moderately satisfied with the prospect of the JOLO blended care app. End-users indicated that they would like to receive information about activity levels and functional status. However, some issues need to be addressed before JOLO can be implemented. The main issues were the accuracy of the activity recognition, the over-generalization of the functional status and the lack of benchmarks and guidelines for healthy mechanical loading in a free-living environment. Moreover, it is necessary that the device can be carried in a more comfortable way to facilitate its integration in everyday life. The results of our study show the potential of JOLO for the assessment of functional status, including joint loading, in people with OA and to inform technology developers on the requirements/future changes that are needed to facilitate the integration of JOLO as a blended care rehabilitation program. This is the first step in obtaining an estimate of the mechanical loading on cartilage in a free-living environment. In future, this information on mechanical loading can be used to inform clinicians and provide patients with personalized feedback on their functional status.

## Figures and Tables

**Figure 1 sensors-20-06967-f001:**
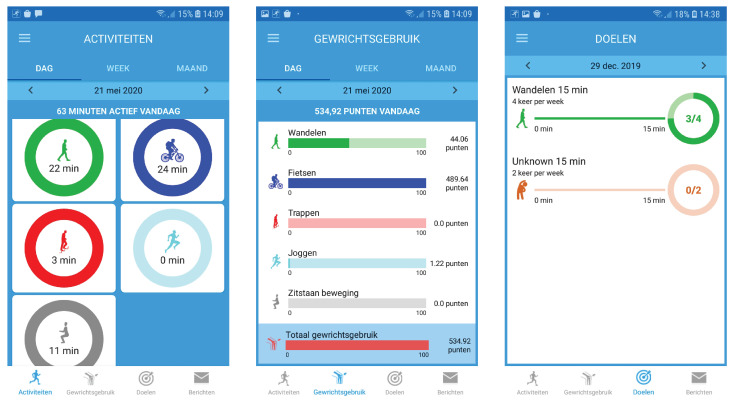
Screenshots of the patient interfaces (from **left** to **right**) activities that can be viewed in a day (=dag), week (=week) and month (=maand) overview. The app provides the total minutes active during that day (= minuten actief vandaag). The functional status (=Gewrichtsgebruik) gives a similar overview as the activities in a day, week and month. The goals (=doelen), in the third panel, show the goals that are set for that week. If a goal has been specified for an activity, the app shows how close the user is to reaching it. In this example, the goal is to walk (=wandelen) for 15 min, four times a week. These interfaces will be used on the smartphone app by the osteoarthritis (OA) patients.

**Figure 2 sensors-20-06967-f002:**
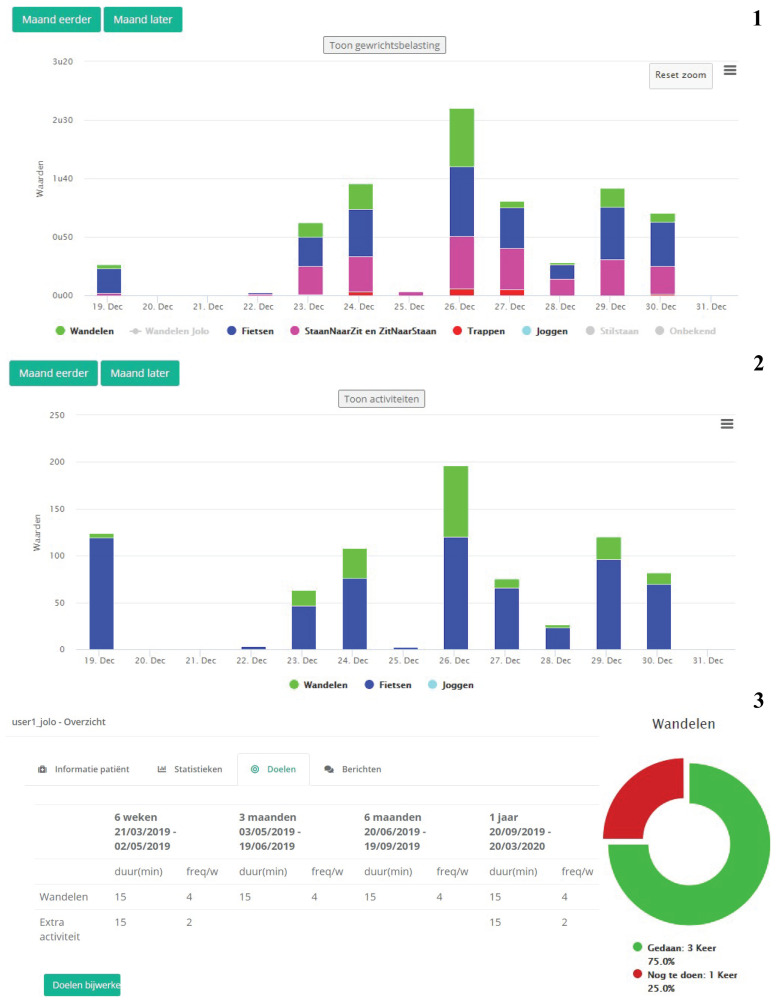
Screenshots of the (1) activities, (2) functional status, and (3) goal screen of the clinical dashboard for the health-care professionals. The activities are displayed in stacked bar graphs per day for walking (=wandelen), cycling (=fietsen), sit-to-stand (ZitNaarStand), stand-to-sit (=StaanNaarZit), and running (=joggen). The second graph shows the corresponding functional status for that time period. The third overview shows the goals (=Doelen) for each time period. The considered time periods are 6 weeks (=weken), 3 months (=maanden), 6 months (=maanden), and 1 year (=jaar).

**Figure 3 sensors-20-06967-f003:**
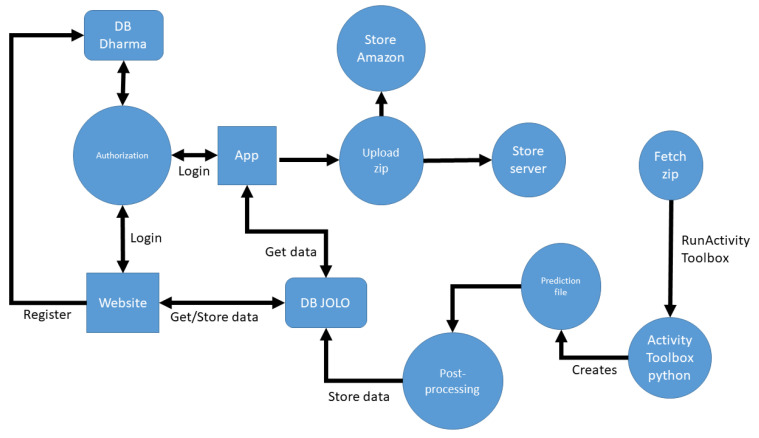
Schematic overview of the JOLO (Joint Load) architecture. The JOLO database (DB JOLO) can be considered as the central component of the architecture. Using API calls, it provides information to both the app and the clinical dashboard. The app regularly uploads data to Amazon S3 and the backend server. Uploaded data is asynchronously fetched, processed by the activity toolbox, postprocessed and stored in the JOLO DB. App and clinical dashboard authorization is managed by a separate database (DB Dharma).

**Figure 4 sensors-20-06967-f004:**
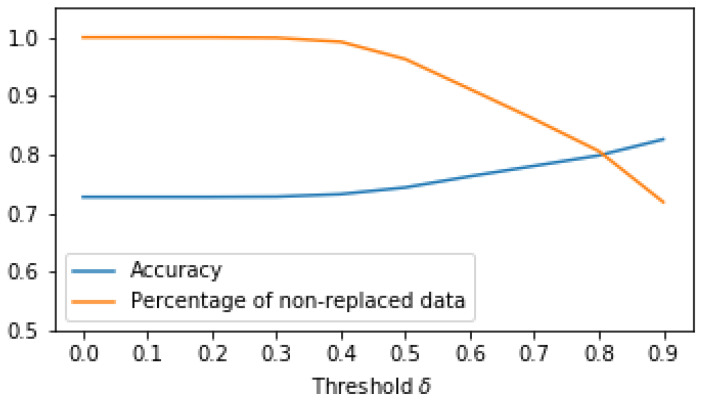
Accuracy and percentage of non-replaced data for different δ thresholds of postprocessing step 1.

**Figure 5 sensors-20-06967-f005:**
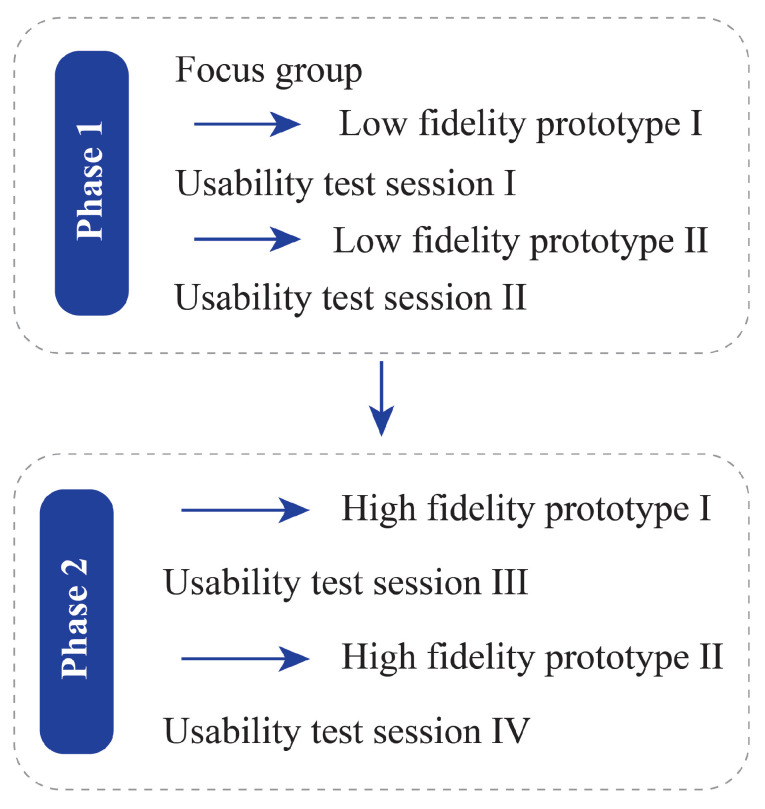
Data collection and JOLO refinement flow divided into two phases, each with two iterations. From each usability test session, invaluable information was derived that was implemented in the design of the prototype that followed it.

**Figure 6 sensors-20-06967-f006:**
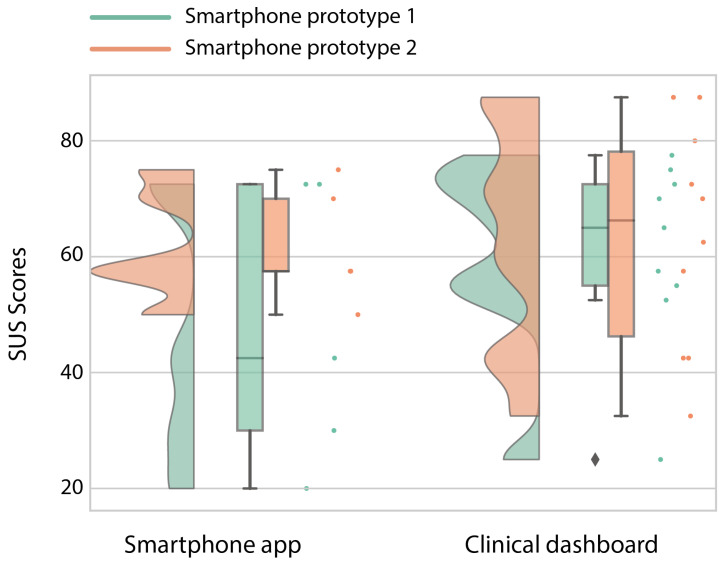
Raincloud plot [42,43] of the System Usability Scale (SUS) for the first and second iteration of the low-fidelity prototype for the smartphone app and the clinical dashboard. The maximum score is 100.

**Figure 7 sensors-20-06967-f007:**
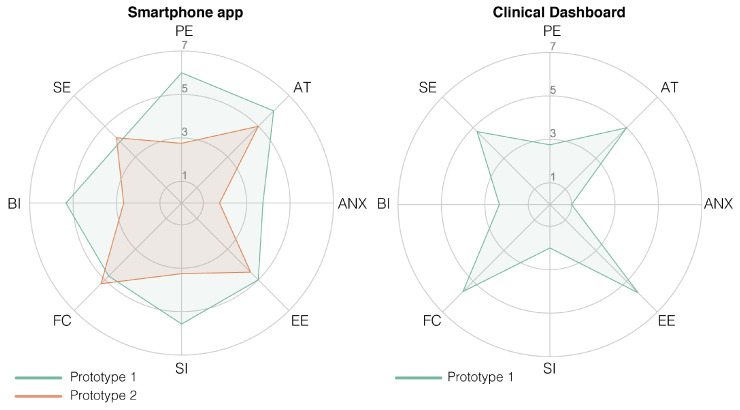
Radar plot of the Unified Theory of Acceptance and Use of Technology (UTAUT) scores of the high-fidelity prototype of the smartphone app (plot on the **left**) and the clinical dashboard (plot on the **right**). A score of 3.5 is the neutral score; thus, everything above the 3.5 is a positive score (except anxiety, which should be below 3.5). The acronyms in the plots are as follows: PE = performance expectancy, AT = attitude towards using technology, ANX = anxiety, EE = effort expectancy, SI = social influence, FC = facilitating conditions, BI = behavioral intent and SE = self-efficacy.

**Figure 8 sensors-20-06967-f008:**
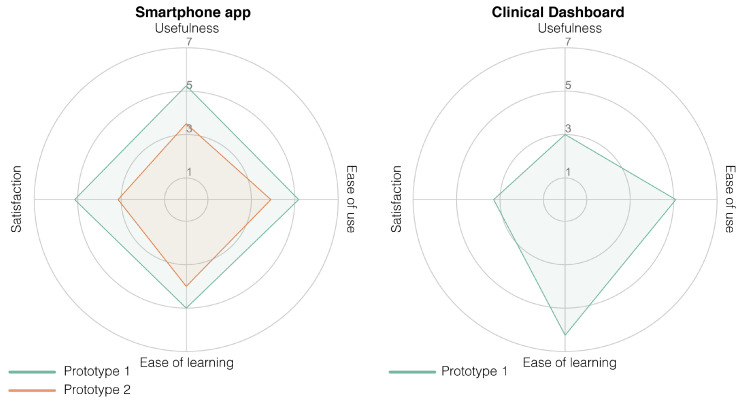
Radar plot of the Ease of Use Questionnaire (USE) scores of the high-fidelity prototype of the smartphone app (plot on the **left**) and the clinical dashboard (plot on the **right**). A score of 3.5 is a neutral score; thus, everything above the 3.5 is considered a positive score.

**Figure 9 sensors-20-06967-f009:**
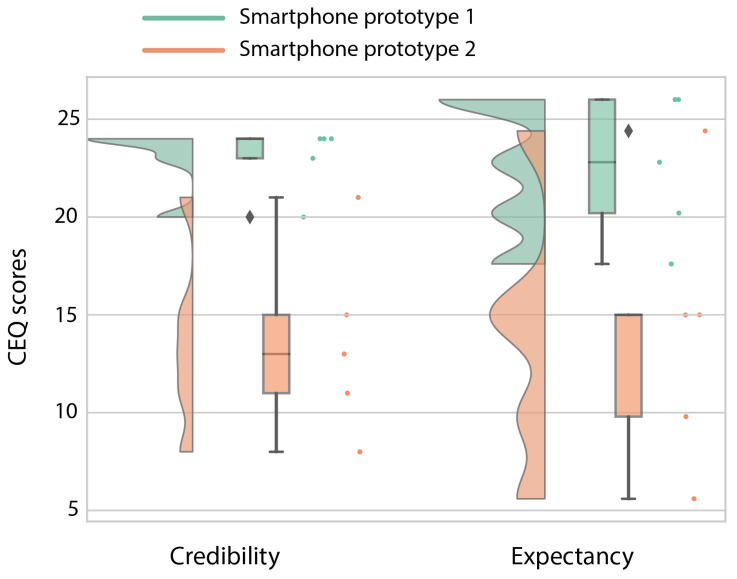
Raincloud plot [42,43] of the Credibility/Expectancy Questionnaire (CEQ) scores of the high-fidelity prototype of the smartphone app. The maximum score of this questionnaire is 27, and everything above 13.5 is considered to be a positive score.

**Table 1 sensors-20-06967-t001:** Duration of the collected data used to train the activity recognition model.

Activity	Duration (s)
Walking	5608
Ascending stairs	564
Descending stairs	420
Sit-to-stand	282
Stand-to-sit	283
Jogging	4344
Cycling	3096

**Table 2 sensors-20-06967-t002:** Confusion matrix. The values are normalized so that each row sums to one. The diagonal, in which the ground truth label equals the predicted label, is highlighted in bold.

Prediction:	Walking	Asc. Stairs	Desc. Stairs	Sit-to-Stand	Stand-to-Sit	Jogging	Cycling
**Ground Truth:**							
Walking	**0.9618**	0.0194	0.0134	0.0000	0.0000	0.0000	0.0054
Asc. stairs	0.2688	**0.6452**	0.0430	0.0000	0.0000	0.0027	0.0403
Desc. stairs	0.1161	0.1000	**0.7258**	0.0000	0.0000	0.0548	0.0032
Sit-to-stand	0.0000	0.0035	0.0071	**0.8901**	0.0922	0.0000	0.0071
Stand-to-sit	0.0035	0.0035	0.0000	0.0813	**0.8975**	0.0000	0.0141
Jogging	0.0266	0.0000	0.0002	0.0002	0.0002	**0.9727**	0.0000
Cycling	0.0271	0.0199	0.0010	0.0003	0.0020	0.0000	**0.9498**

**Table 3 sensors-20-06967-t003:** Average joint contact force impulse expressed in (N/BW)· s per cycle (i.e., stride or repetition) for the hip and knee joint for the different exercises and populations. We used those averages to create the joint loading profiles expressed in points by normalizing the joint loading impulses to the joint loading impulse of healthy walking.

		Joint Loading Impulse (N/BW)· s			Joint Loading Profile (pt)		
		Controls	HipOA	KneeOA	no-OA	HipOA	KneeOA
**Walking**	Hip	192.77	175.74	183.14	1	0.91	0.95
	Knee	140.83	132.36	140.92	1	0.94	1
**Ascending stairs**	Hip	213.34	233.28	274.51	1.11	1.21	1.42
	Knee	229.2	235.43	242.27	1.63	1.67	1.72
**Descending stairs**	Hip	215.26	223.51	290.58	1.12	1.16	1.51
	Knee	218.18	225.02	240.63	1.55	1.6	1.71
**Sit Down**	Hip	136.34	133.68	154.51	0.71	0.69	0.8
	Knee	248.87	240.45	234.37	1.77	1.71	1.66
**Stand Up**	Hip	124.95	111.03	129.74	0.65	0.58	0.67
	Knee	221.54	199.7	199.74	1.57	1.42	1.42

**Table 4 sensors-20-06967-t004:** Participant characteristics of the low-fidelity prototype. N is the number of subjects, and technology usage is assessed on a visual analog scale from 0–10; 0 = “no confidence” to 10 = “extremely confident.”

	Low-Fidelity Prototype Testing			
	Smartphone App		Clinical Dashboard	
	Prototype I	Prototype II	Prototype I	Prototype II
N	5	5	9	10
Physical therapist/surgeon	-	-	5/4	5/5
Age (range)	44–68	49–79	25–60	27–36
Sex (F/M)	3/2	2/3	4/5	2/8
Technology use (range)	5–9	5–10	7–9	7–10

**Table 5 sensors-20-06967-t005:** Participant characteristics of the high-fidelity prototype. N is the number of subjects, and technology usage is assessed on a visual analog scale from 0–10; 0 = “no confidence” to 10 = “extremely confident.”

	High-Fidelity Prototype Testing		
	Smartphone App		Clinical Dashboard
	Prototype I	Prototype II	Prototype I
N	5	5	5
Physical therapist/Surgeon	-	-	5/0
Age (range)	54–68	52–61	27–44
Sex (F/M)	2/3	2/3	2/3
Technology use (range)	6–8	4–10	6–10

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
