# Peer review of "Towards the Monitoring of Functional Status in a Free-Living Environment for People with Hip or Knee Osteoarthritis: Design and Evaluation of the JOLO Blended Care App"

_sensors, 2020, doi:10.3390/s20236967_

Round 1

Reviewer 1 Report

The authors propose the concept and design of a novel blended-care app called JOLO (JointLoad) that combines free-living information on activity with lab-based measures of joint loading that estimates functional status. JOLO aims to provide a better overview of a patient’s functional status and furthermore the patient’s rehabilitation strategies can be further optimized.

This paper presents work in an important field.

To the best of my knowledge, the paper is original and unpublished. The paper is well organized.

In the Introduction section, the paper should be devoted to give a comprehensive review of literature, papers on rehabilitation may also be included. It is recommended to analyze better the literature, more recent articles are available:

Evaluation of a Rehabilitation System for the Elderly in a Day Care Center, Information 2019, 10, 3.

The paper needs to be revised and the English grammar has to be proofread by an English native speaker. References are not correctly reported in the text, reading the text has been difficult.

Reviewer 2 Report

Manuscript titled “Towards monitoring of functional status in a free-living environment for people with hip or knee osteoarthritis: design and evaluation of the JOLO blended care app” deal an important issue of cartilage biology. The authors propose the concept and design of a novel blended-care app called JOLO (Joint Load) that combines free-living information on activity with lab-based measures of joint loading that estimates functional status.

This paper is interesting and well organized. These findings are very important in cartilage biology and in the clinical context. This paper is suitable with the remit and purpose of the journal even if minor concerns need to be solved before suggest publication.

In the introduction section please add a sentence regarding the beneficial and deleterious effects of mechanical loading on cartilage tissue, to strengthen your work and to help better readers understanding. Please quote the following interesting papers:

The Effect of Mechanical Loading on Articular Cartilage. J. Funct. Morphol. Kinesiol. 2016, 1, 154-161. https://doi.org/10.3390/jfmk1020154

Early elbow osteoarthritis in competitive enduro motorcyclist. Scand J Med Sci Sports. 2020 Jul;30(7):1287-1290. doi: 10.1111/sms.13664. PMID: 32246791.

Pag 1, line 21, please quote adequate missing reference as follow:

Unicompartmental Knee Arthroplasty: The Past, Current Controversies, and Future Perspectives. J Knee Surg. 2018 Nov;31(10):992-998. doi: 10.1055/s-0038-1625961. Epub 2018 Mar 7. PMID: 29514367.

Unicompartmental Knee Arthroplasty: Indications, Outcomes, and Complications. Conn Med. 2017 Feb;81(2):87-90. PMID: 29738151.

In the introduction section some sentence are without adequate references, I found only a symbol ?, please provide to fix this problem.

Lines 33-38: This period is unnecessary, since it explains a concept that is largely known.

Line 44: Would you refer to iTunes or App store, considering mobile phones? There are some differences to be aware of:

iTunes - desktop/laptop program for downloading media and mobile apps

iTunes Store - mobile app for downloading media

App Store - mobile app for downloading iOS apps and games

Mac App Store - Mac OS program for downloading apps and games for your Mac OS device(s)

Line 45: “In 2013 there were over 150000 active apps combined in those two app stores”. Could you provide a more recent estimation? App development is constantly growing.

Lines 44-63: This part lengthens too much the introduction and drives the reader out of topic. I suggest to shorten or cut some sentences in order to maintain a solid line of argument. Good, instead, the final part (64-90) in which the reasons behind the study are expressed, simple and clear.

Line 95: Is this app available only in Dutch? Why are the languages in Figure 1 and 2 different? It would be more helpful and reasonable to provide screenshots in English.

Line 132-136: This entire piece of the text is a bit messy and shallow. The message is not straightforward and clean. I suggest revising this part, from a linguistic point of view.

Line 228: could you provide your estimation about resultant joint contact forces of the hip and knee and describe how they correlate with the three JOLO profiles?

Lines 394-401: This part should be moved in discussion as “limitations”

Line 424: Did you try the app in different smartphones to verify if the accelerometer runs well with the app?

Line 516: Change “persons” with “people”

In the conclusion section please highlight better the scientific/clinical relevance of your work, and also please, provide a clear message of the importance of this paper into the scientific community.

Reviewer 3 Report

Very difficult to follow and dense text.

Seems as though this is of less applied use than theoretical interest.

This work is too convoluted and reserach is too preliminary.

Text too long.

Research very limited.

Idea not clearly tested-in the clinical sense.

Maybe present the research separately from the concept and in more basic less overwhelmingly technical form.

Give clinical details, and better theoretical background and basic research underpinning idea.  

Round 2

Reviewer 2 Report

In my opinion the paper now is ready to be publishable.

This manuscript is a resubmission of an earlier submission. The following is a list of the peer review reports and author responses from that submission.

Round 1

Reviewer 1 Report

It is a well-written report on use of cutting edge wearable sensor technology to monitor osteoarthritis in a home setting. Although the results are not "encouraging", it does point in the direction for further development.

Reviewer 2 Report

In this paper, the authors proposed an app for monitoring the functional status in a free-living environment of people suffering from osteoarthritis.

In my opinion, the paper is well written and well presented. Despite the scientific novelty is very limited (since it is basically a system that uses well-known methods) the app seems very functional for the study.

I have a question for the authors. In line 148, they stated that they predict the probability of each of the seven activities.  if the prediction value is belowe 0.5, it is labelled as "unknown activity". Is not 0.5 too low? I think that 0.5 is amenable to a coin toss. Did the authors tried with higher values?

Finally, here are my suggestion for improving the paper:

  • Correct PostGresQL with PostGreSQL;
  • Add some graphics for the obtained results to facilitate the reading and to highlight immediately the goals achieved,

Reviewer 3 Report

The introduction section has several assumptions and unsubstantiated claims. This section must be reviewed thoroughly, removing or editing any statements that cannot be referenced sufficiently. There is no strong motivation for the need of this app, or at least not one I can see while reading this introduction section. Not sure how it fits. The authors must reconsider their angle and way of reporting their motivation and relevant literature.

Significant information is omitted or poorly reported in the technical implementation section of this paper.

The paper is very poorly written, with serious omissions, and overgeneralisations, while in many instances lacking strong scientific merit.

I urge the authors to reconsider their methodological approach to their work and rethink the design of their system as I truly believe such a system can be vastly important within the context of home rehabilitation and virtual therapeutic assistance.

I provide below detailed discussion on my thoughts of the study presented in this paper:

Line 34-35: What does this statement mean? Either clearly explain why and how this is relevant to this paper or remove. The link provided does not work.

Line 36-37: Saying that IMUs as found on most mobile phones is wearable technology is misleading and untrue. Wearables and phones do share some sensors/technologies, but that does not make them by definition “wearables”.

Line 38-39: Are you only considering Belgium? How do these statistics translate outside Belgium?

Line 39-40: “As many people already own a smartphone, it constitutes as an ideal and readily accepted wearable device to monitor physical functionality.” This is a huge, unsubstantiated assumption and a massive leap of faith.

Line 41: “Furthermore, many commercial activity monitoring applications (apps) are available nowadays” Any examples? Do they require you to strap your phone on specific places on your body (e.g. a waist pouch) or do they come in combination with other wearables – i.e. Fitbit, Garmin, Apple Watch?

Line 43-47: Again this is a muddled argument full of unreferenced inaccuracies and assumptions. “only activity recognised is gait” – by whom? Which app/system? What type of gait? Why is stairs and cycling relevant here?

Not sure why the authors disqualified “most” activity recognition apps based on what is presented here. A clearer argument would be that there are no apps that focus on weight bearing. Or that activities that are associated with weight bearing between limbs are not accurate enough, but this needs argument needs to be made in much more rigorous and scholarly fashion.

As it is presented now, it reads very rushed and coming out the authors presumptions and opinions rather than the literature.

Line 47-48: Unclear what this sentence means:  “When wearables and/or technology (i.e. web-based programs) are used in a therapeutic setting, we can speak of e-health”. Also, do you define technology to be anything web-based in this paper? Because a few lines above you defined technology as hardware sensors – IMUs.

Line 55-57: This is a bold statement to make. I urge the authors to either tone it down or provide substantial references to back it up. “However, most use the total amount of physical activity and do not discriminate between the different activities recorded in daily life and thus do not take the mechanical loading (i.e. joint loading) into account.”

Figure 2 should be rotated 90 degrees for clarity. Difficult to read as is.

Line 73: You state that IMU data are collected.

Line 92: You state that only triaxial accelerometer data are recorded. Please be more consistent. If you did not collect data from the whole IMU, and only from its accelerometer, be specific about it.

How did you account for sensor drifts? How did that affect the activity recognition? Did you give your participants any specific instructions – have the phone in their pockets or did you give them special straps? Did the type of clothes a participant wore affect your results (i.e. skinny jeans with tight pockets vs loose sweatpants)? Edit: Line 104 is the first time a hip bag is mentioned!

Line 103: change “machine learned model” to “machine learning model”

Line 107-114: How did the model you created for classification from healthy participants fit to OA participants? Did you at least age match?

Line 115-118: Does that mean you took a ‘Pythagoras theorem’ value from the total o the 3 axis? How did that not completely muddle your data? You essentially converted a triaxial accelerometer to an agnostic one dimension force gauge. What I am concerned with here is that your aim was activity recognition, you started with a device containing an IMU, ignored all gyroscope and magnetometer data (which could at least be used for drift correction and sensor fusion techniques to give more accurate data) and then reduced the dimention of data capture from the accelerometer from 6 (including the negatives) to 1!

Line 118-120: Does that mean you performed activity recognition for every second?

Line 141-142: typo, broken sentence

Line 149: How was the 0.5 value threshold decided? It is not entirely clear this is a valid approach.

Line 152 -153: If training examples from healthy participants were used, how do you know that any changes in recognition accuracy are not caused because by the OA participant condition (or even special circumstances such as age)?

Line 164-167: This is all very poorly explained. If you only used what I assume was a peak detection algorithm to check for peak values on your converted monodirectional data, how do you translate that to actions? I cannot see how a phone mounted on the hip can see clear cycle revolutions for example when cycling, or how it knows if the stand to sit is starting or ending (I assume there is going to be a big spike on either end, and even in the middle if someone stands up too fast). Really sceptical on the accuracy of the data and the way they were collected and analysed.

Still cannot understand why the full IMU and all 3 axis were not used. Also, there is a number of proven ML algorithms the authors could use for activity recognition. Not entirely sure how all the meta processing of the data actually affected them.

Line 165-167: “Find the frequency”[…], “(3) multiply this frequency by the window length to compute the number of steps.” What is meant by the frequency? Peaks per minute? And wat is the window length? Is that the 150 samples? How does that help to compute the number of steps?

This entire section is very muddled and needs serious re-editing for the interest of clarity.

Line 172-195: This is all information regarding another study. Is this study published? Can you provide a reference? This is a lot of information to be reported in this paper if its from another study. The authors must edit this section back and only report the absolute necessary.

Line 196-200: This feels like an overgeneralisation. Do you have any examples of other people doing this? Using kinematic information to generalise to the whole population or do you assume that all people of the same condition have the same walking style? Difference in weight does not mean they all apply the same force to the ground when walking for example.

Line 205: Not sure if I understand this, but is collecting points by excertiong yourself to higher forces the goal of this app? I thought it was all about therapeutic intervention. Here it reads like by going up and down a set of stairs just to collect your points quota is good enough.

Line 205-206: 10,000 is not a metric for good health. It was a marketing campaign by a Japanese firm in 1965. Still I am not sure how this links to the app.

Line 218: citation [20] is a guideline by WHO for the west-pacific population. Is this still applicable for your target population in Belgium (Belgium locale assumed based on statements in the introduction section – Line 38-39)

Section 2.2: Who were the end users that participated? OA? Physios? Both? Demographics?

Did you only have two phases? One low fidelity and one high fidelity? If that was it, its not really enough to call it an iterative design process. You just did focus groups with two prototype evaluations. It may sound more like participatory design or even a technology probe approach?

Line 230-233: Last sentence does not make any sense. Typo? Please review.

Line 234-239 must go at the top of this section for clarity along with some more detailed information on your participants.

Section 2.2 is not clear that it is an intro to 2.2.1 and 2.2.2 and was surprised by the next two subsections. The authors may want to revisit this section and make their presentation style clearer. There is some repeat information between these 3 sections that can be removed with 220-239 acting purely as a signpost to 2.2.1 and 2.2.2.

Line 241-247: I am sure that Lewis (your reference [21]) never supported that 5 participants will definitely give you 85% of the errors. In fact, if you read the citing paper more carefully, you will see that Lewis is providing another point of view directly criticising that statement and that “5 participants = 85% of the problems” is flawed in most situations and only applicable under very specific circumstances.

Line 265: The longitudinal aspect of this study was never mentioned in this paper until now.

Line 281-285: the results section is not the place to describe what you did in this phase and the iterative design procedure you followed, hence my earlier comment on not being an iterative process when only mentioning 2 focus group style phases.

Section 3.1. SUS is not a valid approach when testing a low fidelity prototype. In this stage, it would be more preferable to run other user centered design methodologies (participatory design for example, or a low fidelity interaction study where participants suimulate interactions in a wizard of oz environment) to help you understand what the users need and expect.

Demographics are presented in a very poorly way. Please consider how other publications present their demographic information. Add the scale labels in the table caption (0 ”no confidence at all”, 10 ”extremely confident”).

How was technology confidence used when analysing your findings?

Line 391: “the therapists indicate that the app is usable but currently not useful.” Then why is this paper interesting if it cannot solve the only aim it was set to solve: assist therapists?? What is the contribution other than, as you reported there, a not useful app??

Line 391: No, you did not use the IMU. You only used 1 calculated force dimension of the accelerometer. There are studies that show in detail that IMUs can be used to accurately identify specific kinematic events (e.g. gait cycles including stance/support events).

Line 395-397: “We tried to predict joint loading by only using the IMU sensors in the smartphone to obtain a more personalized method [32]. While this method has a smaller error for gait than using the

 population average, the mean average error is still 16% [32].” Is that a reference to a previous work or this work? This does not add anything and makes your current context more muddled than it currently us. Please consider removing this citation to your previous work.

Line 398-402: The future work proposed does not come from the findings and results of this study as reported on this paper.

Conclusion section is exceptionally short. Even though there is an attempt to summarise their findings, the authors chose to focus on the positive responses and fail to be critical on what they found.

"the healthy (change to health - TYPO) care professionals are moderately satisfied with the prospect of the JOLO blended care app."

Yes but they also could not see how it could be useful.

The authors say what they should do but not how.

This section needs significant work before it can be called a conclusions section.